# ANFluid: Animate Natural Fluid Photos base on Physics-Aware Simulation and Dual-Flow Texture Learning

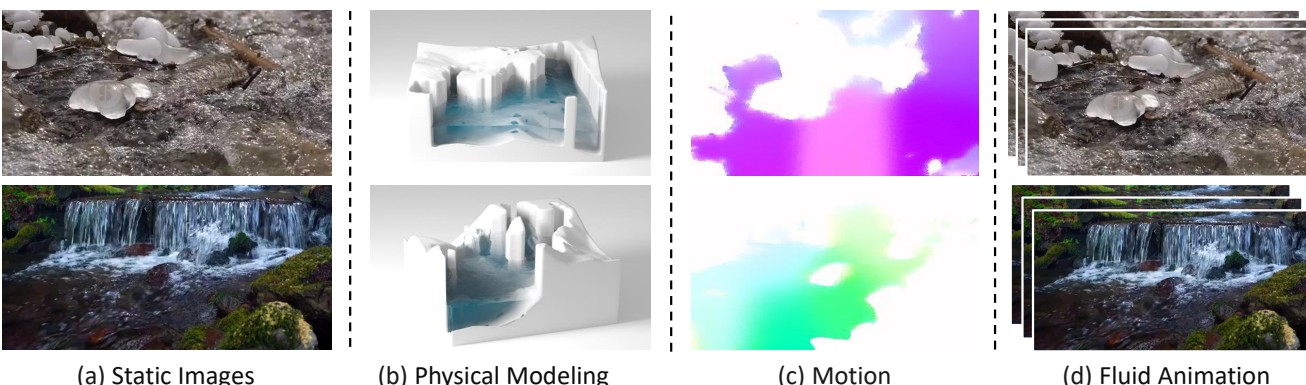

| (a) Static Images | (b) Physical Modeling | (c) Motion | (d) Fluid Animation |

**Figure 1: Motivation of the proposed work. (a) represents the still natural photo with fluid phenomenon, (b) refers to the 3D model from physics simulation, (c) means the estimated motion according to (b), and (d) is the desired dynamic display with (a) as input. From static (a) to dynamic fluid animation (d), a set of motion that conforms to physical motion laws is needed.**

## ABSTRACT

Generating photorealistic animations from a single still photo represents a significant advancement in multimedia editing and artistic creation. While existing AIGC methods have reached milestone successes, they often struggle with maintaining consistency with real-world physical laws, particularly in fluid dynamics. To address this issue, this paper introduces ANFluid, a physics solver and data-driven coupled framework that combines physics-aware simulation (PAS) and dual-flow texture learning (DFTL) to animate natural fluid photos effectively. The PAS component of ANFluid ensures that motion guides adhere to physical laws, and can be automatically tailored with specific numerical solver to meet the diversities of different fluid scenes. Concurrently, DFTL focuses on enhancing texture prediction. It employs bidirectional self-supervised optical flow estimation and multi-scale wrapping to strengthen dynamic relationships and elevate the overall animation quality. Notably, despite being built on a transformer architecture, the innovative encoder-decoder design in DFTL does not increase the parameter count but rather enhances inference efficiency. Extensive quantitative experiments have shown that our ANFluid surpasses most current methods on the Holynski and CLAW datasets. User studies further confirm that animations produced by ANFluid maintain

better physical and content consistency with the real world and the original input, respectively. Moreover, ANFluid supports interactive editing during the simulation process, enriching the animation content and broadening its application potential.

## CCS CONCEPTS

• **Computing methodologies** → **Animation**.

## KEYWORDS

Fluid Animation Generation, Scene-Aware Physics Simulation, Dual-Flow Texture Learning

## 1 INTRODUCTION

Drawing inspiration from the enchanting photographs and newspapers of the Wizarding World in Harry Potter, the transformation of static images into animations represents a captivating and burgeoning field, pivotal to the advancement of multimedia editing. This process seeks to amplify the visual imagination, transcending the mere texture of a solitary image. Diffusion models [13] and advanced generative models [1, 2, 40] have been instrumental in achieving realistic and high-fidelity animations through comprehensive end-to-end learning processes. However, accurately simulating the intricacies of nature phenomena, especially fluid dynamics in the real world, without the direct application of physical laws continues to be a formidable challenge.

For dynamic fluid photo creation, an effective strategy to address these challenges involves the extraction and estimation of motion fields from static images. Holynski and Mahapatra et al. [14, 26] introduced a specialized phase for estimation of the motion field,

which manipulates the deep features extracted to introduce dynamic qualities into static images. Furthermore, an innovative approach within animation generation networks [11] incorporated a physics solver to enhance animation realism by simulating motion. However, these methods typically employ a unified physical solver across various scenarios without fully considering the solver's specific conditions of applicability. This oversight can result in a generalized lack of perception regarding the fluid characteristics relevant to different scenes, thereby compromising the accuracy of the solution. Meanwhile, with respect to texture feature acquisition, these methods enhance animation effects by expanding the parameterization of neural networks, which is offset by the substantial increase in training costs. Moreover, due to inherent design constraints, they do not fully address meticulous feature extraction and effective texture mapping, resulting in consistency imperfections, such as hollow textures and lack of sharpness.

As shown in Figure 1, this study aims to animate natural fluid photos (AnFluid) by introducing Physics-Aware Simulation (PAS) and Dual-Flow Texture Learning (DFTL) methodologies to breathe life into static natural fluid photos. PAS is adept at deriving a physical model (as shown in Figure 1(b)) from the initial inputs, subsequently deducing a plausible motion trajectory (see Figure 1(c)) to serve as the cornerstone for the animation. PAS judiciously selects an appropriate physics solver tailored to each scene, thereby preserving the rich tapestry of physical laws observed in nature. To address common animation pitfalls such as holes and blurriness, DFTL forecasts dynamic textures based on the estimated motion, significantly elevating the animation's quality. Furthermore, our innovative ANFluid framework facilitates interactive editing during simulation, thus enriching the animation content and expanding the scope of the application. This suite of techniques ensures higher fidelity to the physical realities of the real world and greater alignment with the original image's content.

- We propose a fluid short animation creation method from static photos that leverages physics-aware simulation (PAS) to produce more physically realistic fluid dynimics. The physics-guidance approach to motion estimation aligns more closely with the objective laws of physics.
- We design a dual-flow texture learning (DFTL) that effectively mitigates affects such as holes and blurriness textures during the animation generation process. Bidirectional self-supervised optical flow estimation and multi-scale wrapping can strengthen feature detail extraction and dynamic texture association ability.
- We integrate the physics-based and date-driven-based methods within the AnFfluid framework, thus achieve more physically realistic fluid animation generation effects and obtain highly competitive results on the public Holynski and CLAW datasets.

## 2 RELATED WORK

This article discusses the enhancement of still images with animated motions. Initially, Chuang [6] et al. introduced a stochastic motion texture for simple harmonic motion in dynamic picture areas, requiring users to specify motion parameters. Okabe [29] et al. used a fluid video library to synthesize fluid animations from a single image, prompting users to specify motion regions. Machine learning advancements have led to various approaches for fluid animation generation, categorized into direct generation and staged approaches involving feature extraction and evolution for realistic animations.

### 2.1 End-to-End Fluid Animation Generation

In the early stages of research, Yitong Li et al. proposed a framework utilizing a hybrid VAE-GAN [19] end-to-end model to generate videos from text [22]. This framework was designed by integrating three network modules: a conditional keypoint generator, a video generator, and a video discriminator. This design allowed the synthesis from text to video to be initially effective. Chao [4] and Walker [37] focused on improving CNN video generators by focusing on specific functionalities such as human pose generation. However, generating fluid dynamics poses a more complex challenge because of the variability of fluid shapes and motion. Generating videos from text lacks the ability to specify initial scenes, leading to random scene generation based on descriptions. Denton [8] and others proposed deterministic trajectories for dynamic effects and random collisions during motion, combining Fixed Prior (SVG-FP) and Learned Prior (SVG-LP) models. They introduced a recurrent inference network for estimating potential distributions at each time step, showing promising results on the MNIST dataset. However, the effectiveness of this approach in the generation of fluid animations remains unexplored.

With advances in computing power and dataset sizes, AIGC has shown significant progress in video generation. Runway and Open AI have introduced General World Models and Video generation models as world simulators, relying on vast priors learned by large models for predictions. Despite offering control over video effects through input prompts, inherent randomness poses challenges in precise control [10]. This randomness is particularly noticeable in video generation, leading to variations in animated scenes and phenomena such as "unextinguishable candles" and "ghost chairs". These issues stem from the limitations of probability statistics in expressing physical causality, Sora's inability to assess global rationality, and overlooking critical thresholds in physical processes.

In essence, the underlying issue lies in the fact that these methods have not established authentic physical models as the basis for animation generation. The approach proposed in this paper aims to address this fundamental problem by incorporating a more intelligent physics-solving module. This, in turn, enables the efficient generation of physically realistic fluid animations from a single static image.

### 2.2 Staged Fluid Animation Generation

Thi-Ngoc-Hanh [20] divides the entire animation generation process into four steps: extraction of the animation region, flow generation, preservation of curve deformation, and cyclic deformation. However, they did not employ machine learning techniques but used a unified algorithm, similar to the approach used by Yung-Yu Chuang and others [6] in the early stages, to extract flow, generate curve deformations, and cyclic deformations. Holynski [15] introduced the Eulerian motion, a physical model, as a characteristic evolution throughout the generation process. This idea provided

**Figure 2: Method overview. The process begins by generating a depth map and initial motion field from input images and prompts. PAS estimates the motion field, and a multi-scale wrapping image texture feature learning network (MWT) extracts and warps image features based on this field, creating fluid animations iteratively. Additionally, a bidirectional self-supervised optical flow estimation network (BSF) enhances texture feature learning by providing dual-flow constrained motion for MWT during training. In an optical flow diagram (like Init Motion), different colors represent the direction of pixel movement, while the color saturation indicates the speed of motion.**

inspiration for subsequent work by Siming Fan [11], who further developed this concept by replacing the simple Eulerian motion module with Surface-only Fluid Simulation (SFS). Additionally, Fan proposed a Surface-Based Layered Representation (SLR) that complements SFS, enhancing the physical realism of the generation process and the granularity of editing.

Our work integrates the strengths of the aforementioned studies and introduces a novel motion prediction module, an intelligent perception-based physics solving module to improve the performance of physics, and a video generation module based on the transformer architecture, which addresses the challenges of voids in high-velocity scenarios, resulting in further improvements and advancements in the overall fluid animation generation process.

## 3 METHOD

Given a static image containing a liquid region along with prompts, our task is to generate an animated video that dynamically represents the fluid's characteristics through motion effects while ensuring that the flow adheres to physical principles. This paper proposes ANFluid (as shown in Figure 2), a physics solver and data-driven coupled framework that combines physics-aware simulation (PAS) and dual-flow texture learning (DFTL). For the input image and prompts, data preprocessing (DP) is performed to extract the depth and initial motion information. PAS algorithm infers a physically plausible motion field ( Sec. 3.3). On the basis of this motion field and extracted image features, DFTL performs image warping and

decoding to generate fluid animations. The DFTL comprises multi-scale wrapping image texture feature learning network (MWT) ( Sec. 3.2)and bidirectional self-supervised optical flow estimation network (BSF) ( Sec. 3.1). During the training phase, BSF provides MWT with bi-fluid constraint motion information to enhance MWT's ability to extract feature details and texture correlations.

### 3.1 Optical Flow Prediction via Bidirectional Constraint

This paper proposes a self-supervised optical flow estimation network (BSF) based on the advanced network architecture proposed by [34], which incorporates bidirectional constraints. This aims to address the aforementioned limitations and meet the requirements of fluid optical flow estimation more effectively to bi-fluid constraint motion information for MWT. The detailed structure of the network is shown in Figure 3. This structure mainly draws on the network architecture proposed in recent advanced work [34], and in order to further enhance the network's accurate estimation of the transparent fluid region, bi-directional sequences are used for constraint, strengthening the network's grasp of subtle details.

During the training phase of the model, given the lack of high quality ground truth optical flow data in our dataset, we have incorporated the principles of unsupervised optical flow estimation training methodologies [24, 27, 35] and incorporated commonly employed loss functions. To further enhance the performance of the model, we have integrated several unsupervised components

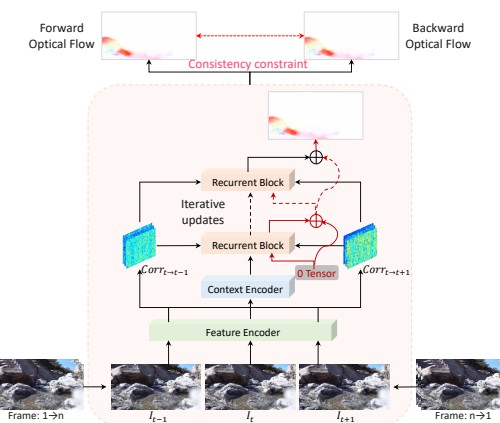

Forward Optical Flow

Backward Optical Flow

Consistency constraint

Recurrent Block

Iterative updates

Recurrent Block

$Corr_{t\rightarrow t-1}$

Context Encoder

0 Tensor

$Corr_{t\rightarrow t+1}$

Feature Encoder

Frame: 1→n        $I_{t-1}$        $I_t$        $I_{t+1}$        Frame: n→1

**Figure 3: Bidirectional self-Supervised optical flow estimation (BSF). In training, we input bidirectional video sequences into the optical flow estimation network to generate optical flow for both forward and backward sequences, enforcing consistency constraints between them.**

that have exhibited efficacy in previous research, including the loss of smoothness Ls [38], the loss of census Lc [27], and the loss of distortion of the boundary expansion [25]. Moreover, to better capture the effects of bidirectional sequence constraints, we have incorporated a constraint loss, formulated as described in Eq. 1:

$$ConsistencyLoss = \frac{1}{N}\sum_{i=1}^{N}|\sqrt{x_f^2 + y_f^2} - \sqrt{x_b^2 + y_b^2}|, \quad (1)$$

where $N$ represents the total number of pixels, $x_f, y_f$ represent the $x$ and $y$ directional optical flow values obtained from the forward sequence, and $x_b, y_b$ represent the $x$ and $y$ directional optical flow values obtained from the backward sequence. This loss function further imposes bidirectional consistency constraints at the pixel level, enhancing the model's ability to estimate complex motions in bidirectional optical flow estimation, thereby improving estimation accuracy.

## 3.2 Texture Feature Learning Utilizing Transformer

To efficiently extract image features and maintain their continuity, we utilized the Transformer structure [36], which has been proven effective in a wide range of generative tasks [9, 18, 43, 44]. It performs well in tasks that require warping operations on features. Our multi-scale wrapping image texture feature learning network (MWT) of dual-flow texture learning (DFTL) incorporates the essence of the Transformer structure from Swin-Unet [3]. We employ the U-Net [31] architecture commonly used in generative models [5, 30, 33, 39], adopting a multiscale skip connection approach to enhance the detailed texture features. For the extracted features, moderate distortion of the image features is applied based on the estimated motion field ($M$) to obtain the features of the moved image. A detailed explanation of the estimation of the motion field ($M$) will be provided in Sec. 3.3, briefly mentioned here. This process can be formalized as shown in Eq. 2:

$$I_f(T_0 \rightarrow T_i) = D(E(I_f(T_0)) \circ M_i(x, y)) \quad (2)$$

where $I_f(T_0)$ represents the first frame of input image, $M_i(x, y)$ represents the motion field for a pixel, where i denotes the ith frame. $I_f(T_0 \rightarrow T_i)$ represents the animation generation process from the 0th frame to the ith frame. $E(*)$ and $D(*)$ represent the encoder process and decoder process, respectively.

Ultimately, the wrapped image features are processed through the decoder to generate a frame in the animation. Repeating this process yields the complete fluid animation. To generate higher quality animations, we adopt a method similar to [11], considering $T_i$ frames as a linear combination of $T_0$ and $T_n$ frames for training and using $T_0$ frames instead of $T_n$ frames during testing to generate loop animations. In practical implementation, to maintain the smoothness and continuity of the generated animation texture, we utilize softmax splatting [28] and learnable composition factors $Z(T_0)$ and $Z(T_n)$ to determine the contribution of overlapping pixels to the transformation of $I_f(T_0)$ and $I_f(T_n)$, as proven in [11, 26]. Unlike other methods, we emphasize the importance of $Z(*)$ in image synthesis, which affects the synthesis results. Therefore, based on the feature encoder mentioned above, we construct an independent Z channel for learning $Z(*)$, with $Z(*)$ learning being influenced by the feature encoder but not causing a reciprocal effect.

We train the animation generation network using the loss function as shown in Eq. 3:

$$\begin{aligned} L_{\text{image}} = &\left| I(T_i) - I_{\text{gt}}(T_i) \right| \\ &+ \lambda_0 \|\text{VGG}(I(T_i)) - \text{VGG}(I_{\text{gt}}(T_i))\| \\ &+ \lambda_1 \text{Disc}(I(T_i)) \\ &+ \lambda_2 (L_{\text{local}}^{\text{p}} + L_{\text{global}}^{\text{p}}), \end{aligned} \quad (3)$$

where $I(T_i)$ is the generated frame image at time frame $i$, $I_{\text{gt}}(T_i)$ is the ground truth frame image at time frame $i$. $\lambda_0, \lambda_1$, and $\lambda_2$ are weighting parameters. During training, our aim is to optimize the quality of generated images while considering human perception. We use constraint techniques from previous research, including pixel-level constraints, perceptual loss, and discrimination based on image authenticity. The neural network of the transformer architecture shows a smoother continuity of feature extraction. We introduce a transformer-based MAE network for the perceptual loss function, which has been shown to be effective [23]. We input the generated image ($I_{\text{recon}}$), and the reference image ($I_{\text{ref}}$), into a pre-trained MAE model to extract the representations. Then, we define local perceptual loss as the Euclidean distance between the feature representations Eq. 4.

$$L_{\text{local}}^{\text{p}} = \|F^l(I_{\text{recon}}) - F^l(I_{\text{ref}})\|_1. \quad (4)$$

In Eq. 4, $F^l$ denotes the MAE backbone, which produces representations in set $\{T^l, Q^l, K^l, V^l\}$. Unlike CNN, the shadow layers of the transformer tend to capture local semantic information, while the deeper layers favor to present the global semantic information. We refer to the specific implementation of [23].

By implementing the comparison of the feature distributions between the generated images and the reference images based on the optimal transport theory Eq. 5 proposed in [23]:

$$L_{\text{global}}^{\text{p}} = \sum_{i=1,\ldots,n,CLS} W_p^p \left( F_i^l(I_{\text{recon}}), F_i^l(I_{\text{ref}}) \right), \quad (5)$$

where $F_i^l(I_{\text{recon}})$ is one of the extracted features from $F^l(I_{\text{recon}})$ and similarly for $F^l(I_{\text{ref}})$. Computing the Wasserstein distance for all the labeled images, we can obtain the distribution-aware loss as the sum of the Wasserstein distances $W_p^p(u, v)$ on $F_i^l(I_{\text{recon}})$ and $F_i^l(I_{\text{ref}})$.

## 3.3 Motion Field Estimation with Physics-based Solver

In Sec. 3.2, we present a comprehensive elucidation of the operational mechanics of animation generation models. However, we omitted a comprehensive elucidation of the motion-field estimation. In this section, we will expound upon our motion field estimation methodology, which imbues a physics-aware simulation. This approach encompasses the generation of initial motion fields through interactions, the incorporation of scene-aware physics-based solving techniques, and motion field smoothing methodologies while taking into account the scene complexity.

*3.3.1 Initial motion of interactive sparse labels.* To allow users control over the final generated effects, we refer to the methods employed by [11, 26]. This enables users to specify dynamic components while providing discrete motion directions that do not exceed 10. We utilized nearest-neighbor averaging to determine the initial velocity for all pixels within the liquid region. Specifically, the velocity at pixel $(i, j)$ is the exponential average of adjacent pixels.

$$v_{i,j} = \frac{\sum_k V_k e^{-d(k,ij)^2/\sigma^2}}{\sum_k e^{-d(k,ij)^2/\sigma^2}}, \quad (6)$$

where $v_{i,j}$ represents the velocity of the sequentially numbered pixel points in the fluid region. $V_k$ denotes the k-th labeled velocity, $d(k, ij)$ represents the Euclidean distance between the pixel $(i, j)$ in the image and the position of the k-th label. $\sigma$ is a parameter related to the size of the image.

*3.3.2 Introducing scene-aware fluid simulation.* As stated in Sec. 3.2 previously, the primary task for static image input is scene categorization to facilitate the selection of an appropriate physical solver. In this regard, we propose a joint integration of the scene perception network with the encoder component of the animation generation network, as depicted in Figure 2. To achieve this, we enhance the existing encoder architecture by introducing a classification head. Moreover, we enforce constraints on the classification head using the cross-entropy loss function, which takes the form as described in Eq. 7:

$$\text{CrossEntropy}(y, \hat{y}) = - \sum_i y_i \log(\hat{y}_i), \quad (7)$$

where y represents the true labels and $\hat{y}$ represents the predicted labels. During the training process, we incorporate this classification task into the constraint loss function of the animation generation network, resulting in the final formulation of the loss function as presented in Eq. 8:

$$L = L_{\text{image}} + \lambda_3 \text{CrossEntropy}(y, \hat{y}), \quad (8)$$

where $\lambda_3$ is weighting parameter.

For different scenes, we have adopted two physical solvers, namely the particle-in-cell (PIC) method and the Shallow Water Equation (SWE) method. These solvers are chosen based on different characteristics of the scene, such as waterfalls, rivers, and springs.

**PIC** [12, 45] is a physics simulation method that combines both Eulerian and Lagrangian perspectives. We initialize it using a particle perspective while selecting a standard cubic grid as the framework for the grid perspective. The specific process is as follows.

We use a monocular depth estimation network to obtain the depth and position information of the particles. This process can be described as follows (Eq. 9):

$$\begin{aligned} x &= (u - c_x)/f_x \cdot d, \\ y &= -(v - c_y)/f_y \cdot d, \\ z &= d, \end{aligned} \quad (9)$$

where $x$, $y$, $z$ represent the extracted particle position, $d$ is the depth, $u$, $v$ are the pixel coordinates on the image, and the camera parameters are set to $f_x, f_y, c_x, c_y$, corresponding to a perspective camera with a field of view (FOV) angle of 90 degrees in height.

The 3D velocity and position information of particles required for PIC simulation, as well as boundary information, are obtained through predefined camera poses, user input prompts, and 2D velocity transformations. The specific method can be referred to [11] (Eq. 10):

$$\frac{d}{dt} \begin{bmatrix} u \\ v \end{bmatrix} = \begin{bmatrix} f_x & 0 \\ 0 & f_y \end{bmatrix} \begin{bmatrix} 1/z & 0 & -x/z^2 \\ 0 & 1/z & -y/z^2 \end{bmatrix} \begin{bmatrix} \frac{dx}{dt} \\ \frac{dy}{dt} \\ \frac{dz}{dt} \end{bmatrix}, \quad (10)$$

where $f_x, f_y$ are the camera's intrinsic parameters, $u, v$ are velocities on the projected image, and $x, y, z$ are 3D positions in the scene.

Once all initialization data are obtained, we use a Taichi [16, 41, 42] implementation of a mixed Eulerian-Lagrangian method [7, 21, 45] to simulate the subsequent motion of each particle. The motion equations are as follows (Eq. 11):

$$\begin{aligned} \frac{\partial u}{\partial t} + u \cdot \nabla u &= -\frac{1}{\rho} \nabla p + g, \\ \nabla \cdot u &= 0, \end{aligned} \quad (11)$$

where $u$ represents fluid velocity, $\rho$ represents density, $p$ represents internal pressure of the fluid, and $g$ represents external forces, considering only gravity.

Finally, we perform an inverse operation using the above equations to obtain the 2D motion field required by the network.

**SWE** [32] find extensive applications in domains such as oceanic currents and atmospheric circulation. They represent a specialized expression of the Navier-Stokes equations, assuming a scenario where the depth of the fluid is significantly smaller than the lateral dimensions, which naturally is a 3D simulation based on 2D information. Specifically, SWE requires information such as water surface height, water surface velocity, and boundary information, they all correspond well to the parameters in the equation. Furthermore, this reduction in dimensionality leads to a concomitant

decrease in computational complexity. The conservative form of these equations is expressed as follows:

$$\frac{\partial(\rho\eta)}{\partial t} + \frac{\partial(\rho\eta u)}{\partial x} + \frac{\partial(\rho\eta v)}{\partial y} = 0, \qquad (12)$$

$$\frac{\partial(\rho\eta u)}{\partial t} + \frac{\partial}{\partial x}\left(\rho\eta u^2 + \frac{1}{2}\rho g\eta^2\right) + \frac{\partial(\rho\eta uv)}{\partial y} = 0, \qquad (13)$$

$$\frac{\partial(\rho\eta v)}{\partial t} + \frac{\partial}{\partial y}\left(\rho\eta v^2 + \frac{1}{2}\rho g\eta^2\right) + \frac{\partial(\rho\eta uv)}{\partial x} = 0, \qquad (14)$$

where $\eta$ is the total fluid column height (instantaneous fluid depth as a function of $x$, $y$ and $t$), and the 2D vector $(u, v)$ is the fluid's horizontal flow velocity, averaged across the vertical column. Further $g$ is acceleration due to gravity and $\rho$ is the fluid density.

Similarly to the PIC method, we obtain the water surface height through the estimation of the monocular depth and use the method described in Sec. 3.3.1 to obtain the water surface velocity. Obstacle information is obtained through user input prompts, and boundary conditions are set as reflective boundaries to ensure the free flow of water into and out of the boundaries. As the water surface height information obtained from the monocular depth estimation is represented by values in the range [0,255], the values obtained for the water surface flow velocity under the user's guidance are aligned with this magnitude. In the simulator, we standardize the units to meters. We have employed the open-source solver Anuga to solve the shallow-water equations, configuring it with the aforementioned information to simulate the velocity field after a certain period.

*3.3.3 Motion Field Smoothing.* Through the aforementioned physical evolution process, we obtained a sequence of velocity fields that evolve over time. However, the simulation-generated results only account for the velocity on the fluid surface, neglecting variations in fluid thickness [11]. As a result, the outcomes are fragile when dealing with complex initial states. Therefore, we need additional approaches to compensate for the detailed motion of fluid textures. Following the baseline approach, we employed a convolutional network to transform the simulated motion field into a more detailed and realistic motion. This network was trained for a conventional-style transfer task, with L2 loss between the output velocity and the ground truth. For specific configurations, refer to the pipeline described in [17].

## 4 EVALUATION

In this section, we compare our proposed DFTL ( Sec. 4.1) against state-of-the-art learning-based fluid animation methods on the Holynski and CLAW datasets to virify its effectiveness. We then discuss the effects of different motion field estimates on the quality of fluid animation, including our proposed scene-aware physics-based solver ( Sec. 4.2).

### 4.1 Fluid Video Quality

**Dataset and Evaluation Metrics.** For evaluation, we used Holynski data set validation and the CLAW dataset to assess the performance of the animation generation network. It is important to note that in this section the motion of all the methods compared is guided by the optical flow of the entire video to differentiate

the performance of synthesis techniques. For quantitative results, we first utilized the 60th frame of each sequence and compared the metrics for all intermediate frames. We used PSNR to show the overall average error, SSIM to show errors in regions with a significant amount of texture, and LPIPS (Alexnet version) to display perceptual loss. This setup is consistent with that of previous work [11].

**Quantitative Comparison.** Table. 1 presents quantitative results that compare our method with previous approaches. The Reproduced Holynski approach is a typical method based on single-layer learning, which globally animates scenes. The SLR approach is a method that animates scenes using a surface-based layered representation. Quantitative evaluation is carried out on Holynski's validation set [14]and the CLAW test set [11], focusing on the first 60 frames. The *"fluid region"* refers to the static background region replaced by input images during metric computation to improve quality. The baselines are compared under the same ground truth motion, following the settings of previous work [11]. The metrics of other works are taken from previous research [11]. We can observe three main findings from Table 1: (1) Our method outperforms previous work in most terms of metrics on the CLAW dataset, indicating its superior performance in generating outdoor fluid animations, particularly in complex scene animation tasks. (2) Specifically considering the LPIPS metric, our method surpasses previous work on both the CLAW Testset and the Holynski Common Validation Set, highlighting its ability to generate fluid animations that align better with human perception and provide a more impressive visual experience. (3) Our method, compared to the baseline SLR model, achieves outstanding performance with fewer model parameters and surpasses prior research in multiple metrics.

**Quantitative Ablation.** Table. 2 presents the results of our ablation experiments, where SLR serves as the baseline method. Ours (Non-Zlayer) indicates the use of a Transformer architecture but without employing a separate Z-channel. "Ours" (Non-Multi-scale warp) incorporates a transformer architecture, a separate Z-channel without multi-scale warping. Our model, denoted as "Ours", incorporates a transformer architecture, a separate Z-channel, multiscale warping, and an MAE based perceptual loss. Compared to the SLR method, our approach "Ours" has shown significant improvement in the LPIPS metric and achieved excellent results in other metrics. This suggests that the addition of a separate Z-channel plays a crucial role in the linear combination of features between the initial frame ($T_0$) and the target frame ($T_n$) during the synthesis process. This enhancement improves the accuracy of combining features from preceding and subsequent frames, thereby further enhancing the generation quality. Using multi-scale warping enhances the learning of fine-grained features, leading to comprehensive improvements in all metrics. Building upon this, we incorporated MAEloss into our method, further enhancing its performance on the LPIPS metric, surpassing all previous research efforts.

**Qualitative Comparison.** Figure 4 illustrates the visual detail comparison among Runway[1], baseline, and our method. In Figure 4 (a), the indicated portion shows that our method has improved the problem of holes present at the beginning of the study. In Figure 4

---

[1]We utilized the same functionality as in Runway (i. e., providing a single image, prompts to generate videos), and provided consistent inputs to serve as end-to-end comparison examples.

**Table 1: Quantitative comparison.**

| Dataset | Methods | All Region | | | Fluid Region | | | Params |
|---|---|---|---|---|---|---|---|---|
| | | LPIPS | PSNR | SSIM | LPIPS | PSNR | SSIM | |
| Holynski Common Validation Set | Reproduced Holynski | 0.0798 | 25.03 | 0.7787 | 0.0657 | 25.88 | 0.8007 | - |
| | Modified Holynski | 0.0793 | 24.75 | 0.7758 | 0.0656 | 25.72 | 0.8000 | - |
| | SLR (Baseline) | 0.0834 | **25.14** | **0.7795** | 0.0657 | 26.10 | 0.8030 | 16.39 MB |
| | Ours | **0.0791** | 24.68 | 0.7694 | **0.0615** | **26.31** | **0.8046** | 8.09 MB |
| CLAW Testset | Reproduced Holynski | 0.2067 | 20.26 | 0.5955 | 0.2029 | 20.36 | 0.5961 | - |
| | Modified Holynski | 0.2078 | 19.97 | 0.5923 | 0.2041 | 20.10 | 0.5934 | - |
| | SLR (Baseline) | 0.2040 | 20.79 | **0.6080** | 0.1975 | 20.80 | **0.6077** | 16.39 MB |
| | Ours | **0.1572** | **21.13** | 0.6030 | **0.1600** | **21.10** | 0.6021 | 8.09 MB |

**Table 2: Quantitative ablation**

| Dataset | Methods | All Region | | | Fluid Region | | |
|---|---|---|---|---|---|---|---|
| | | LPIPS | PSNR | SSIM | LPIPS | PSNR | SSIM |
| Holynski Common Validation Set | SLR (Baseline) | 0.0834 | **25.14** | **0.7795** | 0.0657 | 26.10 | 0.8030 |
| | Ours (Non-Zlayer) | 0.0872 | 24.54 | 0.7737 | 0.0710 | 25.65 | 0.7982 |
| | Ours (Non-Multi-scale warp) | 0.0857 | 24.64 | 0.7732 | 0.0682 | 25.87 | 0.8012 |
| | Ours | **0.0791** | 24.68 | 0.7694 | **0.0615** | **26.31** | **0.8046** |
| CLAW Testset | SLR (Baseline) | 0.2040 | 20.79 | **0.6080** | 0.1975 | 20.80 | **0.6077** |
| | Ours (Non-Zlayer) | 0.2104 | 20.26 | 0.5947 | 0.2061 | 20.36 | 0.5957 |
| | Ours (Non-Multi-scale warp) | 0.2052 | 20.36 | 0.5887 | 0.2031 | 20.44 | 0.5897 |
| | Ours | **0.1572** | **21.13** | 0.6030 | **0.1600** | **21.10** | 0.6021 |

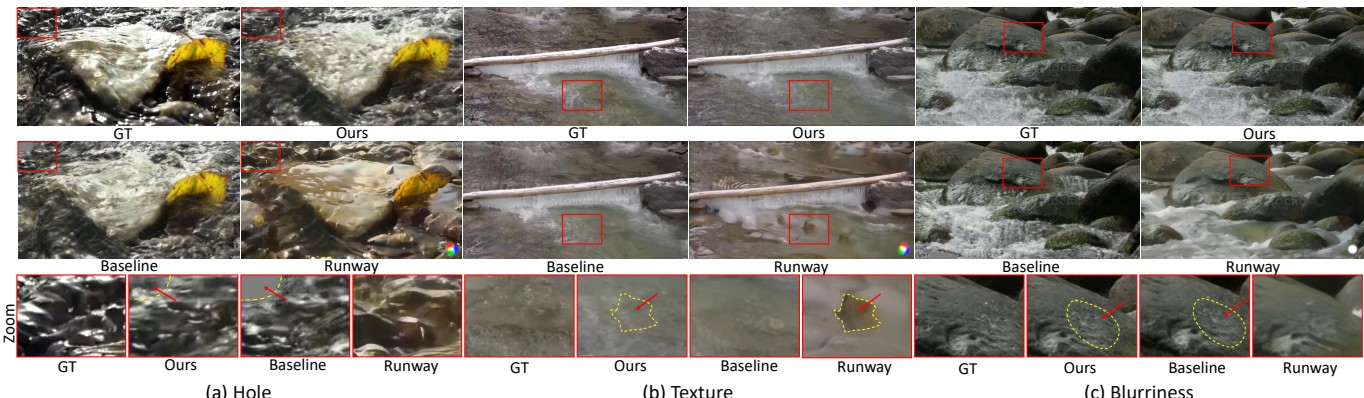

**Figure 4: Qualitative comparison with runway and baseline. Demonstrating superiority in generating fluid animation effects, this method alleviates issues in three aspects: (a) Hole, (b) Unrealistic texture, and (c) Blurriness.**

(b), the indicated portion demonstrates that our method does not produce erroneous textures like that of Runway. In Figure 4 (c), the indicated portion shows that the clarity of the images generated by our method is higher than the baseline.

**User Study.** In order to clarify the strengths and weaknesses of our method in real-world applications, we conducted a serious user study, inviting participants to subjectively evaluate fluid videos generated by our model. Three methods were evaluated in user studies: SLR-SFS used in the baseline, Runway, and our proposed

method. To visually present the performance of each method in various aspects, we designed five evaluation metrics: video quality, picture integrity, fluid motion authenticity, editing comprehension ability, and texture realism. The detailed explanation of each item can be found in our supplemental appendix.

In the study, we presented participants with obfuscated optical flow images as input and contrasted these with fluid animation outcomes that were synthesized using the three specified methods. The participants were then asked to evaluate and rank the quality of the

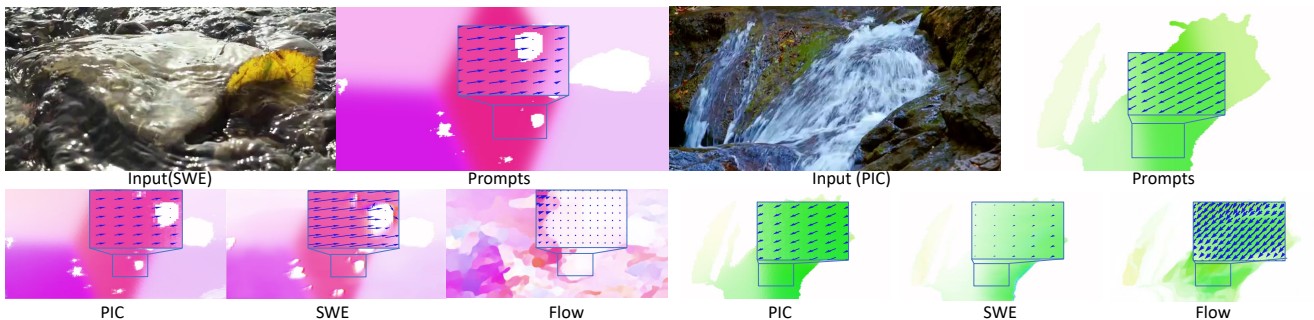

**Figure 5: Motion estimation comparison. It demonstrates the different optical flow results obtained using various motion estimators when perceiving scenes as SWE and as PIC inputs, with brief annotations indicating motion directions.**

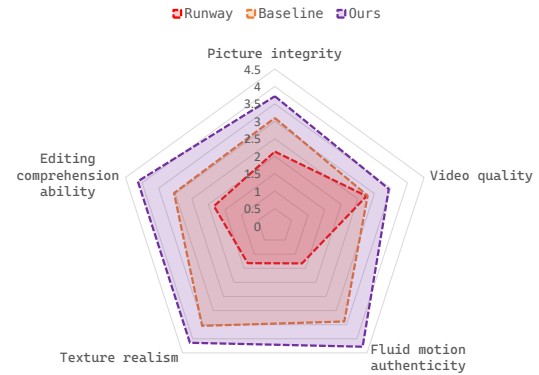

**Figure 6: User study. The subjective ratings are derived from a subset of samples from the CLAW dataset, covering a balanced representation of both PIC and SWE scenes.**

results along a specific dimension across varying scenarios. We aggregated data from a total of 170 completed questionnaires. Figure 6 shows the average scores obtained by converting the rankings into scores based on weighted ranking. Our approach has demonstrated multifaceted enhancements. Notably, there has been a significant boost in the perceived credibility of fluid motion and overall image coherence over the baseline, underscoring the efficacy of the PAS and DFTL techniques. Additionally, unexpected improvements were also observed in video quality and editing comprehension. On the contrary, the animations produced by Runway were found to lag in all evaluated dimensions, with the authenticity of textures being particularly inferior. Such shortcomings stem from the marked deviation of the generative models from authentic textures, yielding a sub-par visual experience. These findings accentuate the merits of our staged animation generation approach.

### 4.2 Different Motion Estimations

Figure 5 presents the results obtained from different motion generation pipelines. All the results from the following motion estimates share the same input, derived from the algorithmically assigned velocity for each pixel point based on user input. We present the results of our study, which includes the comparison of motion maps generated by SWE and PIC methods in different environments.

Additionally, we explore the use of neural networks for motion estimation without physics simulation to highlight differences.

From Figure 5, we derive two important conclusions: Firstly, both PIC and SWE outcomes, due to their reliance on motion-field interpolation, exhibit greater continuity in their results, as exemplified by the image on the left. This stands in contrast to end-to-end methodologies that generate discontinuous velocities, diverging from the natural continuity of water surface movements. Secondly, while PIC typically provides a correctly perceived overall motion, it is prone to loss in accuracy. SWE, on the other hand, consistently produces more coherent and finer motion fields, particularly favoring contexts conducive to shallow-water equations. In the cases of significant elevation differences, the PIC approach outperforms SWE, which has difficulty accurately capturing water flow. For example, utilizing SWE in high-drop scenarios, as shown in the right scene, would neglect the critical vertical descent of waterfalls, recommending the PIC's use in such instances for more verisimilar animations. The contrast in performance between these two simulators endorses our proposed intelligent perception physical-solving module.

## 5 CONCLUSION

This paper introduces ANFluid, an innovative framework that synergizes a physics solver with a data-driven approach, integrating physics-aware simulation with empirical learning to positively animate natural fluid imagery. Unlike previous research, our method capitalizes on PAS to deduce motion fields, which aligns the resultant animations more rigorously with physical laws. DFTL harnesses the power of bidirectional self-supervised optical flow estimation coupled with multi-scale warping to bolster dynamic correspondences, thereby improving the quality of the final animations. There is hope for our work to advance the creation of dynamic fluid photo animation, transitioning from the still photos to the more complex task of dynamic short video generation. With a commitment to fostering collaborative research in this field, we will open-source our code to empower the broader research community to contribute and propel the progress in this area.

Notwithstanding the use of a competitive transformer architecture in the generation process, a degree of blurriness is occasionally observed in the output. In anticipation of this, future work will explore the adoption of more refined network architectures with the aim of further advancing the standard of animation generation.

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
