# OpenReview forum: "$ANFluid: Animate Natural Fluid Photos base on Physics-Aware Simulation and Dual-Flow Texture Learning$"
_acmmm.org/ACMMM/2024/Conference — MM2024 Poster_

### Official Review · Reviewer_v85L · 2024-05-20

**Rating:** 3
**Confidence:** 2

**Summary:**

The paper introduces ANFluid as a framework to animate natural fluid photos by combining physics-aware simulation (PAS) and dual-flow texture learning (DFTL). The PAS component aims to make motion guides adhere to physical laws and can automatically choose suitable numerical solver for different fluid scenes. Meanwhile, the DFTL component enhances texture prediction by employing bidirectional self-supervised optical flow estimation and multi-scale wrapping. The DFTL design uses less parameters and improves inference efficiency. Experiments are conducted on the Holynski and CLAW datasets and show that ANFluid outperforms prior work.

**Strengths:**

For fluid region, the proposed method generally has better performance than SLR.

**Limitations:**

1. The paper is not well-organized and challenging to follow.
2. Many components of the proposed method are adapted from "Simulating Fluids in Real-World Still Images", and the key contribution of this paper is unclear.
3. For the Holynski dataset in Table 1, the proposed method has worse PSNR and SSIM than SLR when considering all regions.

**Suitability:**

2

---

### Official Review · Reviewer_TeLw · 2024-05-22

**Rating:** 4
**Confidence:** 1

**Summary:**

This paper introduces ANFluid, a physics solver and data-driven coupled framework that combines physics-aware simulation (PAS) and dual-flow texture learning (DFTL) to animate natural fluid photos effectively. The PAS ensures that motion guides adhere to physical laws, and DFTL focuses on enhancing texture prediction. ANFluid surpasses most current methods on the Holynski and CLAW datasets.

However, the novelty of this paper is limited.

**Strengths:**

- High-Quality Results:
- -  This paper generates high-quality visual results.
- Writing
- - The writing of this paper is easy to follow.

**Limitations:**

- Dependency on initial inputs:
- - The animation's accuracy depends on the quality of the initial depth and motion field estimations, which might be challenging to obtain in some cases.

- Novelty：
- - The novelty of this paper is limited; the self-supervised optical flow estimation network (BSF) is based on [34]. This will affect the novelty of this paper.
- - The optical flow estimation is also based on the previous approach [24, 27, 35]. The losses include the loss of smoothness Ls [38], the loss of census Lc [27], and the loss of distortion of the boundary expansion [25].
- - The transformer structure is borrowed from [9, 18, 43, 44].
- - The multi-scale wrapping image texture feature learning network is from Swin-Unet [3].
- - The Initial motion of interactive sparse labels is generated by [11, 26].
- - The Motion Field Smoothing is from [17].

- Related works:
- - Some related methods are worth deep discussion:

- - - Zeng, B., Liu, X., Gao, S., Liu, B., Li, H., Liu, J., & Zhang, B. (2023). Face animation with an attribute-guided diffusion model. In Proceedings of the IEEE/CVF Conference on Computer Vision and Pattern Recognition (pp. 628-637).

- - - Shen, S., Zhao, W., Meng, Z., Li, W., Zhu, Z., Zhou, J., & Lu, J. (2023). Difftalk: Crafting diffusion models for generalized audio-driven portraits animation. In Proceedings of the IEEE/CVF Conference on Computer Vision and Pattern Recognition (pp. 1982-1991).


- Reproducibility:
 - - It could be difficult to reproduce this paper since many configurations are not well mentioned or referenced to another paper.
- Training data
- - The training relies on high-quality trianing data.

**Suitability:**

2

---

### Official Review · Reviewer_drf7 · 2024-05-25

**Rating:** 4
**Confidence:** 1

**Summary:**

The authors address the task of image animation for natural fluid photos with physics awareness. They propose AnFluid, an integrated system combining both physics-based and data-driven methods. Specifically, they leverage the physics-aware simulation (PAS) to obtain motion guidance, which is better aligned with the laws of physics. They also design a dual-flow texture learning method to mitigate the artifacts existing in previous methods, such as holes and blurriness. The proposed system is evaluated and compared with existing state-of-the-art competitors and outperforms them in both quantitative and qualitative measurements.

**Strengths:**

1. The paper presented demonstrates good clarity in writing, and the approach is convincing and sound.
2. The application of PAS allows for the selection of an appropriate physics solver tailored to each scene during inference, which can enhance the accuracy of the animation.
3. The integration of data-driven-based training and physics-based inference (in terms of motion field generation) is interesting and valuable for the fluid image animation field.

**Limitations:**

1. During training, AnFluid utilizes self-supervised optical flow estimation (BSF) for motion field generation and warps the intermediate features. However, during inference, motion fields are obtained based on physics simulation models. How does the model address this mismatch in performance, and does it impact the model's effectiveness?
2. From my perspective, although the results generated by AnFluid appear more "physically" accurate, they seem less realistic compared to flow-based or surface-based methods. How is the realism of these results evaluated?
3. The proposed approach relies on the performance of the motion estimation module. It may not be feasible for users to manually select different physics models for motion estimation. I am curious if, in general, the physics-based method outperforms flow-based methods in terms of the final animation results.
4. The authors should discuss some of the limitations of the proposed method.
Typos:
L162: date -> data
L627: virify -> verify

**Suitability:**

3

---

### Official Review · Reviewer_G9Uy · 2024-05-25

**Rating:** 2
**Confidence:** 4

**Summary:**

This paper presents an innovative approach combining physics-aware simulation and data-driven learning to animate natural fluid images. While the work demonstrates some degree of novelty, the presentation and clarity require significant improvement.

**Strengths:**

A novel approach combining physics-based and data-driven methods.

**Limitations:**

The writing is often unclear, and the proposed methods are not described comprehensively, leading to a poor reading experience. Here are specific comments and suggestions:

1. Related Work (Section 2.2): The statement, "However, they did not employ machine learning techniques but used a unified algorithm," suggests a misunderstanding by the authors. Not every research needs to employ machine learning techniques. The authors should acknowledge that alternative approaches can be equally valid and effective.

2. Figure 3: The pipeline of the BSF (Bidirectional Self-supervised Optical Flow Estimation) is not clearly depicted. A more detailed and step-by-step explanation is needed to understand the workflow.

3. Section 3.2: The term "(M)" is mentioned, but it is not explained or illustrated in Figure 2. The authors should clarify what "(M)" refers to and ensure it is properly integrated into the diagram.

4. Iterative Animation Generation (Figure 2): The method described involves iteratively generating multiple frames, but Figure 2 does not clearly illustrate the iterative process. The flowchart is confusing and needs to be reorganized to better reflect the iterative steps.

5. Design of MWT: The rationale behind the design of the MWT (Multi-Scale Wrapping Image Texture Feature Learning Network) is not well-explained. Figure 2 provides limited information, and further elaboration on the design choices and their justifications is required.

6. Baseline Model (SLR): The paper uses SLR as the baseline model. The authors need to justify why SLR was chosen as the baseline and how it compares to other potential baseline models.

7. "Modified Holynski": The term "Modified Holynski" is used but not explained. The authors should provide a clear definition and description of what modifications were made to the original Holynski method.

8. User Study: The user study appears to have a high degree of subjectivity. The necessity and validity of such a subjective evaluation should be thoroughly discussed. Additionally, more objective metrics and evaluations could strengthen the study.

Overall, while the paper introduces some novel concepts, the presentation needs significant revision for clarity and comprehensiveness. The authors should address the above points to improve the readability and robustness of their work.

**Suitability:**

2

---

### Meta-Review · Area_Chair_7dMZ · 2024-06-27

**Recommendation:** Accept (Poster)
**Confidence:** 3

**Metareview:**

This paper received a mixed set of reviews. The reviewer who gave the most negative review was mainly concerned with the writing of this paper. Another two reviewers were concerned with the lack of novelty of this paper as the proposed method adopts many of the components from existing papers. After the rebuttal, one of these reviewers was convinced by the rebuttal and switched to a positive recommendation. Another positive reviewer found that the proposed method of combining physics-based and data-driven methods is interesting. Most of the reviewers find that the results are good. Given these, we'd like to recommend the acceptance of this paper and encourage the authors to carefully revise the paper to incorporate the reviews.